# Influence of social and physical environmental variation on antipredator behavior in mixed-species parid flocks

Colton B. Adams [1,2], Monica Papeş [1,3], Charles A. Price [3], Todd M. Freeberg [1,2]*

1 Department of Ecology and Evolutionary Biology, University of Tennessee, Knoxville, TN, United States of America, 2 Department of Psychology, University of Tennessee, Knoxville, TN, United States of America, 3 National Institute for Mathematical and Biological Synthesis, University of Tennessee, Knoxville, TN, United States of America

* tfreeber@utk.edu

**Data Availability Statement:** All relevant data are within the paper and its Supporting Information files.

## Abstract

Carolina chickadees (*Poecile carolinensis*) and tufted titmice (*Baeolophus bicolor*) regularly form flocks with multiple species through the winter months, including white-breasted nuthatches (*Sitta carolinensis*). Earlier studies found that behavior of both chickadees and titmice was sensitive to mixed-species flock composition. Little is known about the influence of background noise level and vegetation density on the antipredator behaviors of individuals within these flocks, however. We tested for the effects of vegetation density, traffic noise, and flock composition (conspecific number, flock diversity, and flock size) on antipredator behavioral responses following an alarm call playback (Study 1) and an owl model presentation (Study 2) at feeders. We recorded background traffic noise and performed lidar scans to quantify vegetation density at each site. After a feeder had been stocked with seed and a flock was present, we recorded calls produced, and we identified flock composition metrics. We coded seed-taking latency, call latency, mob latency, and mob duration following the respective stimulus presentation and tested for effects of flock composition metrics, vegetation density, and background noise on these responses. For the alarm call playback study, flock composition drove behaviors in chickadees and titmice, and vegetation density drove behaviors in chickadees and nuthatches. For the owl model study, conspecific number predicted behavior in chickadees, and mob duration was predicted by nuthatch number. The results reveal individual sensitivity to group composition in anti-predatory and foraging behavior in simulated risky contexts. Additionally, our data suggest that the modality of perceived simulated risk (acoustic vs. visual) and the density of vegetation influence behavior in these groups.

## Introduction

Individuals often gain numerous benefits from living in groups [1–3], especially in the context of enhanced foraging [4, 5] and avoiding predation [6, 7]. The efficiency of locating and exploiting food sources [8, 9], as well as detecting and responding to predators [10, 11], is

**Funding:** The author(s) received no specific funding for this work.

**Competing interests:** The authors have declared that no competing interests exist.

often increased for individuals participating in group living. Vocalizations are crucial to maintaining group connectivity and cohesiveness in many species [12, 13], with food recruitment calls and antipredator alarm calls being appropriately responded to by both conspecifics and heterospecifics [14, 15]. As proportions of conspecifics and heterospecifics naturally vary within groups, it is necessary to study how group composition may influence foraging and calling behaviors of individuals during potentially risky environmental contexts.

Birds in the family Paridae–the parids (chickadees, tits, and titmice)–routinely form overwintering mixed-species flocks that vary in size and species composition across the northern hemisphere [16]. In the southeastern United States, mixed-species flocks of Carolina chickadees (*Poecile carolinensis*), tufted titmice (*Baeolophus bicolor*), and white-breasted nuthatches (*Sitta carolinensis*) regularly form in the early fall months and remain together into the following spring [17–19]. Chickadees and titmice are core members of these flocks, while nuthatches and other species are common satellite members [20]. During winter months, food resources become scarce, and the physical environment becomes more open from deciduous trees shedding their leaves, raising the risk of predation from visual exposure [17]. Two principal benefits that members in this mixed-species flock system receive are an increased ability to find food and to detect predators [21, 22]. Individual behavior in these flocks is often influenced by birds' calling behavior [13, 23]. In particular, the *chick-a-dee* call produced by chickadees and titmice (and tits in Africa, Asia, and Europe) is a ubiquitous, multi-purpose signal utilized by species in the Paridae and other bird families [13]. Furthermore, individual behavior–including calling, foraging, and antipredator behavior–is sensitive to the proportions of conspecifics and heterospecifics within these flocks [23, 24].

Anthropogenic noise also impacts foraging and antipredator behavior across a variety of animal taxa [25], especially those heavily reliant on acoustic signaling channels for communication [26]. Parids, including chickadees and titmice, are sensitive to levels of noise produced by humans, as evidenced by the responses of individuals to changes in sound intensity [27, 28]. Another component of the physical environment that influences behavior of mixed-species flock members is habitat openness (i.e., a basic measurement of vegetation density in a given area) [29, 30]. However, the majority of studies to date on effects of habitat variability on behavior measure habitat openness on a broad, categorical scale. The relationship between vegetation density, as measured using quantitative tools such as lidar (light detection and ranging) technology, and animal behavior and communication is understudied. Thus, it is important that we seek to quantify this variation with technological approaches that add rigor to quantitative measures of habitat structure. Given that vegetation density naturally varies across the range of habitats in which mixed-species flocks are found, it is reasonable to assume that this variability may play a role in parid antipredator behaviors.

Furthermore, predation risk can be perceived by prey individuals through different sensory channels [31]. The modality (e.g., visual, acoustic, tactile, chemical) in which a predator's signals or cues are detected influences subsequent antipredator behaviors [32–34]. Acoustic and visual predation cues differ in the antipredator responses they elicit from prey individuals across taxa [35]. The speed with which an antipredator response is elicited and the duration that the response period lasts are often determined by the modality of the perceived predation risk and the concurrent risk assessment by individuals within mixed-species groups [36].

We conducted two different field experiments to simulate predation risk near feeders for mixed-species flocks of chickadees, titmice, and nuthatches at an oak-chestnut forest site in eastern Tennessee, U.S.A. The first experiment was a playback of alarm calls originally obtained from titmice. The second experiment used an eastern screech owl (*Megascops asio*) model presentation. The alarm calls of titmice are known to elicit reliable antipredator responses in our mixed-species flock system [37, 38], and the eastern screech owl is well-

documented to prey upon the bird species in our flocks [39]. Although screech owls are primarily nocturnal predators and we carried out our studies during daylight hours, birds in mixed-species flocks with parids respond strongly to visual and acoustic stimuli of screech owls during the day [21, 27]. Typical antipredator responses include fleeing the feeder at the onset of the playback, freezing once perched, and halting seed-taking and calling for a certain period of time. Eventually, individuals will resume calling and seed-taking, and it is also common for flocks to mob at the feeders [40]. In other feeder studies, boldness has commonly been assessed by observing an individual bird's propensity to call and its propensity to return to feeders and resume normal seed-taking after stimulus presentations [40–42]. Mobbing is an antipredator response that likely indicates boldness, riskiness, and aggression [21, 30, 43]. Quicker calling and resumption of feeding after a potential predator has been detected are typically associated with bolder individuals and flocks, as are quicker mob onsets and longer mob durations.

Following the presentation of our respective stimuli, we measured seed-taking latency and calling latency for chickadees, titmice, and nuthatches, and we measured mob latency and mob duration for entire flocks. We tested for the effects of the following environmental metrics on our measured response variables: conspecific number, flock diversity, flock size, traffic noise, and vegetation density (each described below). We formulated five hypotheses to test in this study:

- H1: Seed-taking and call latency measures will decrease across our three focal species with increasing conspecific numbers. We based this hypothesis on the assumption that the number of conspecifics in a flock is the key variable influencing individual anti-predatory behavior. Individuals that are more similar to, and more familiar with, others in the flock are more likely to behave in a bolder manner under conditions of risk [44, 45].

- H2: Increasing flock diversity will determine an increase in individual bird seed-taking and call latencies, an increase in flock mob latency, and a decrease in flock mob duration. Antipredatory behavior may be influenced by flock diversity as individuals that are less similar to, and less familiar with, others in the flock are less likely to behave in a bold manner under a risky context [21, 46].

- H3: As total flock size increases, seed-taking and call latency measures will decrease across our three focal species, flock mob latency will decrease, and flock mob duration will increase. Total flock size may be a key determinant of individual anti-predatory behavior because more individuals within a flock is likely to result in more rapid antipredator behavior and a more rapid return to seed-taking, as mobbing becomes more effective with greater overall numbers of individuals [47, 48].

- H4: Higher levels of noise will increase individual bird seed-taking and call latencies, increase flock mob latency, and decrease flock mob duration. If traffic noise is a main factor in anti-predatory behavior, at louder sites, vigilance and antipredator behaviors should occur with greater delay, with overall noise levels decreasing mob effectiveness [26, 27].

- H5: Individual seed-taking and call latencies as well as flock mob latency will be longer, and flock mob duration will be shorter, at denser vegetation sites. Assuming that habitat density around the feeders is a key variable impacting anti-predatory behavior, increasing vegetation density should be associated with an increase in obscurity and a decrease in conspicuousness, resulting in a slower return to foraging due to increased uncertainty of potential predator location, especially in the alarm call playback study [49, 50].

## Methods

### Study system

We studied the antipredator behavior of mixed-species flocks of Carolina chickadees, tufted titmice, and white-breasted nuthatches at the University of Tennessee Forest Resources, AgResearch, and Education Center (UTFRREC; 36.11˚ N, 84.20˚ W) in eastern Tennessee. Data were collected from late November 2021 to late March 2022 between 8:00 and 17:00 EST. Overwintering mixed-species flocks typically form and persist throughout these months [17–19]. We conducted experiments at 36 feeder sites, which were separated by a minimum distance of 375 meters (m) to ensure that each feeder site was independently sampled and in a different flock's territory [22]. That earlier study [22] involved 15 sites with most of the birds at those sites color-marked and individually identifiable; birds were rarely found at a different feeder than the one at which they were banded. Feeder studies have regularly been carried out at UTFRREC for over 15 years [41, 51–53]. This UTFRREC location is part of the mixed oak-chestnut forests of the southern hardwood forest region of U.S.A., and it is dominated by mixed tree species, including tuliptree (*Liriodendron tulipifera*), hickories (*Carya* spp.), oaks (*Quercus* spp.), and pines (*Pinus* spp.) [54].

Each site consisted of a feeding station that was made of a wooden platform tray (25 centimeters (cm)× 40 cm × 2 cm) attached to a steel pole. The pole was driven into the ground until the platform was roughly 1.5 m from ground level. Beginning in early October 2021 and continuing through the end of this study in late March 2022, the 36 feeding stations were stocked every 10 to 14 days with 50 grams (g) of bird seed to attract flocks. We used a black oil sunflower seed/safflower seed mix that reliably attracts our three focal species, as well as others [51]. Platforms were stocked with 50 g of seed the day of trials. Once a flock was actively exploiting a site (i.e., present and regularly taking seed from the feeding station), we set up a Sennheiser ME-64 electret recording microphone clipped to a WindTech desktop microphone stand and connected to a Marantz PMD-660 recorder (16-bit and 44.1 kilohertz (kHz) sampling rate) with a 15 m auxiliary cord. Recordings were saved as uncompressed WAV files. The recording microphone was angled directly towards the platform and was positioned roughly 1 m away, while the observer (C.B.A.) was positioned at least 10 m away from the feeding station and partially obscured by vegetation [22]. After the recording set up was complete and the observer was in position, we began our trial, which was split into a pre- and post-stimulus presentation period. This experimental set up applied to both our alarm call playback and predator model presentation studies.

### General experimental protocol

During the pre-stimulus period for both of our experiments, we estimated the total number of individuals for each species within the flock. This was done by noting the maximum number for each species that we could detect visually and acoustically in the area of the feeder during a trial period [51], which has been shown to be positively correlated with the actual number of individuals of each species present [22]. Flocks in our eastern Tennessee population typically comprise 2–4 chickadees, 2–4 titmice, and 1–2 nuthatches. Calling behavior was obtained by our recorder for both the pre-stimulus and post-stimulus periods of our trials. We verbally noted when a seed was taken off of the platform by a given species throughout the trial. Observer vocalization at this distance is not shown to significantly influence the behavior of individuals within this mixed-species flock system [22, 27]. Post-stimulus, we noted the seed-taking and calling latencies for the first individual of each species within a flock to take a seed and to call. Additionally, mobbing latency and mobbing duration were measured post-

stimulus. We defined mobbing as a sequence of *D* notes and *chick-a-dee* calls (chickadees and titmice) and/or *quank* notes (nuthatches) produced in rapid succession and lasting for a bout of more than 10 seconds (sec). These notes and calls are used by these birds in a wide range of social contexts, including in antipredator behavior. Once an additional 10 sec had passed with no rapid call production, a mob was marked as concluded. Analysis of calls was done in Cool Edit Pro 2.0 (Syntrillium Software, Scottsdale, Arizona).

**Alarm call playback.** We conducted playback trials using alarm calls previously recorded from tufted titmice (S1 Fig; obtained from Dr. Kathryn E. Sieving). This experiment was run from late November 2021 to late January 2022. Each playback trial included a 60 sec bout of alarm calls played back at a volume of roughly 75 **decibels** (dB) at 1 m, as measured by a Quest Technologies 2100 Sound Level Meter set at C weighting and slow (1 sec) response. The playback speaker was positioned next to the observer (C.B.A.) roughly 10 m away from the feeder. The WAV file was played back from an iPhone 11 connected via Bluetooth to an iHome iBT34 speaker, which was held by the observer during the playback period. Each trial was 30 minutes (min) long. After setting up our equipment, we waited for the first 14.5 min of each trial before the alarm call presentation. The alarm call was played back from 14.5 min to 15.5 min of the entire 30 min experiment period. We stayed in our observing position for the remaining 14.5 min. If a flock had a species where no individual took a seed during the post-stimulus period, a value of 870 (time, 14.5 min, in sec) was entered into our data sheet.

**Predator model presentation.** A realistic plastic model of an eastern screech owl was used in our predator presentation experiment (S1 File; Safari Ltd., 2013, Miami, Florida). This predator is known to prey upon members of the flocks we study [39]. This experiment was run from late January 2022 to late March 2022. Immediately after beginning a trial, the observer (C.B.A.) would walk to the feeding station from their recording position 10 m away from the feeding station, touch the platform with their hand, and then return to their observation space. This served as a control for any effect our subsequent walk to the feeding station to place the model on it might have on the flock's behavior. Each trial was 20 min long. After setting up our equipment, the observer waited for the first 9 min and 55 sec of each trial before the owl model presentation. Following this time passage, the observer walked to the feeder to place the owl model on it. The observer then returned immediately to the previous observation space and stayed in that position for the remaining 9 min and 55 sec. We ensured that the owl was placed on a corner of the feeding station with its head and body oriented towards the center of the platform where the seed was located. We also ensured that all remaining seed was in front of the owl to minimize the chance that a bird would sneak behind the model, avoiding the model's head and eyes, and take a seed [42, 55]. For this experiment, we observed no seed-taking following the presentation of the owl model. Therefore, it was not included as a measured variable in our statistical analyses.

## Background noise and lidar scan analysis

We measured anthropogenic noise levels at each site from November 2021 through March 2022. We collected background noise measurements at all 36 sites using a Quest Technology 2100 Sound Level Meter set at C weighting and slow (1 sec) response to obtain SPL (dB) measures 4 to 6 times at each site. Excluding Sundays, we would record on days between 10:00 and 13:00 EST. At UTFRREC, main sources of background noise came from vehicle traffic on Highway 62 and Union Valley Road. Background noise has been shown to significantly affect the anti-predatory behaviors of individuals in the mixed-species flocks we study [27, 56]. Anthropogenic noise varied widely across sites, with average noise levels ranging from 50 dB to 70 dB (S2 Fig). There was also variation in how variable noise levels were at sites across sampling days. For our data analyses, we used median noise levels.

Using a FARO Focus S 350 HDR lidar scanner, we collected a terrestrial lidar scan at each of the 36 sites from late January 2022 to late March 2022. Each of our sites is surrounded by a distinct composition and density of vegetation, and previous studies suggest that vegetation density may influence anti-predatory behaviors in our mixed-species flock system [30]. For instance, individuals within flocks were more likely to approach and elicit mobbing calls at playback sites of eastern screech owl calls in a dense habitat (forest/shrub) than in an open habitat (field). To collect the lidar point clouds for each site, we erected the scanner, supported by a large tripod, directly above the feeder platform. Each scan was for a full 360 degrees at a maximum distance of 50 m, and scan time per site was roughly 8 min. All scans were processed in FARO SCENE software (FARO Technologies, Lake Mary, Florida) and saved as LAS files for additional analysis. We further processed each file in CloudCompare (V2 2.13.alpha, Open Source Project), a 3D point cloud and mesh processing software. To clean each cloud, we cropped it to remove outliers, and we applied statistical outlier removal (SOR) and noise removal using default settings. We also applied a cloth simulation filter (CSF) to each scan in order to separate ground from non-ground points (vegetation). Cleaned scans (i.e., those that were filtered and included only off-ground points) were saved again as LAS files. These files were then batch-processed using a script written in MATLAB R2023a (MathWorks Inc., Natick, Massachusetts) to generate statistical and topological descriptors for each cloud based on the concept of alpha shapes. Alpha shapes are geometric objects created by connecting points that fall within a certain radius of one another with lines [57]. In 3D spaces like lidar point clouds, this process creates surfaces and volumes which can be compared across sites. We created an alpha shape for each individual point cloud (i.e., each feeder site) using standard algorithms available within the MATLAB programming environment (alphaShape), utilizing an alpha radius of 1 m. The surface area and volume of each resulting alpha shape were then recorded.

## Data analyses

For both experiments, to test our hypotheses (H1-H5) we ran basic linear regression models. Since we had a relatively small sample size for assessing our five social and physical environmental variables, we did not conduct stepwise regression and so only report the full models [see 58]. Our response variables were seed-taking latency, calling latency, mob latency, and mob duration. Explanatory variables for the full models for each species included the number of each focal bird species (chickadees, titmice, and nuthatches), flock size and diversity, background noise (median dB level), and vegetation density (forest alpha volume, in $m^3$). The inverse Simpson index [59] was used to assess flock diversity, and it was calculated as $[(P_{chickadees})^2 + (P_{titmice})^2 + (P_{nuthatches})^2 + (P_{species"x"})^2 + (P_{species"y"})^2 \ldots]^{-1}$, where P is the proportion of each flock composed of chickadees, titmice, nuthatches, and each additional species, respectively [51]. We estimated effect sizes using $r^2$ values obtained through the correlation of variables with the dependent measures. All statistical analyses were performed using IBM SPSS Statistics Version 27.

Previous studies have suggested that predation risk assessment is a function of cue modality, with anti-predatory behaviors differing with respect to whether visual or acoustic signaling pathways are used [27, 36, 60]. Given the parallel nature of our two studies, with one presenting an acoustic cue of predation risk and the other a visual cue, we also ran cross-comparison tests between our experiments. For each of our three focal species, we compared average calling latency, and, for flocks, we compared mob latency and mob duration, as a function of predation risk cue modality (alarm/acoustic vs. owl/visual) using paired t-tests in IBM SPSS Statistics Version 27. These two field experiments were conducted under approved University of Tennessee Institutional Animal Care and Use Committee protocol #1248.

## Results

### Alarm call playback study

We observed a total of 131 Carolina chickadees, 127 tufted titmice, and 53 white-breasted nuthatches across our 36 study sites. Carolina chickadees and tufted titmice were present across all 36 sites, whereas white-breasted nuthatches were present at 29 sites. Flocks averaged 3.64 ± 0.16 (mean ± SE) Carolina chickadees, 3.53 ± 0.26 tufted titmice, and 1.47 ± 0.14 white-breasted nuthatches. The other species observed at one or more sites included red-bellied woodpeckers (*Melanerpes carolinus*; 11 individuals across 10 sites), downy woodpeckers (*Dryobates pubescens*; 7 across 7 sites), Northern cardinals (*Cardinalis cardinalis*; 3 across 2 sites), and Carolina wrens (*Thryothorus ludovicianus*; 5 across 3 sites). The total number of birds observed was 337, the total flock size averaged 9.36 ± 0.35 individuals, and the average flock diversity index was 2.81 ± 0.76.

**Chickadee seed-taking and calling latency.**   We detected a significant effect of conspecific number on post-stimulus seed-taking latency for chickadees ($\beta$ = -0.392, t = -2.444, p = 0.021, $r^2$ = 0.190). An individual chickadee was more likely to come back to the feeder sooner after the playback had ended if there were more chickadees in the flock (Fig 1, left panel). Additionally, we detected a significant effect of forest alpha volume on post-stimulus seed-taking latency for chickadees ($\beta$ = 0.330, t = 2.103, p = 0.044, $r^2$ = 0.141). An individual chickadee was more likely to take a first seed with a longer delay after the playback had ended if there was greater vegetation density around the feeder (Fig 1, right panel). We could not detect an effect of flock size ($\beta$ = -0.144, t = -0.706, p = 0.486), flock diversity ($\beta$ = 0.099, t = 0.486, p = 0.631), or traffic noise ($\beta$ = 0.044, t = 0.275, p = 0.785) on chickadee seed-taking latency.

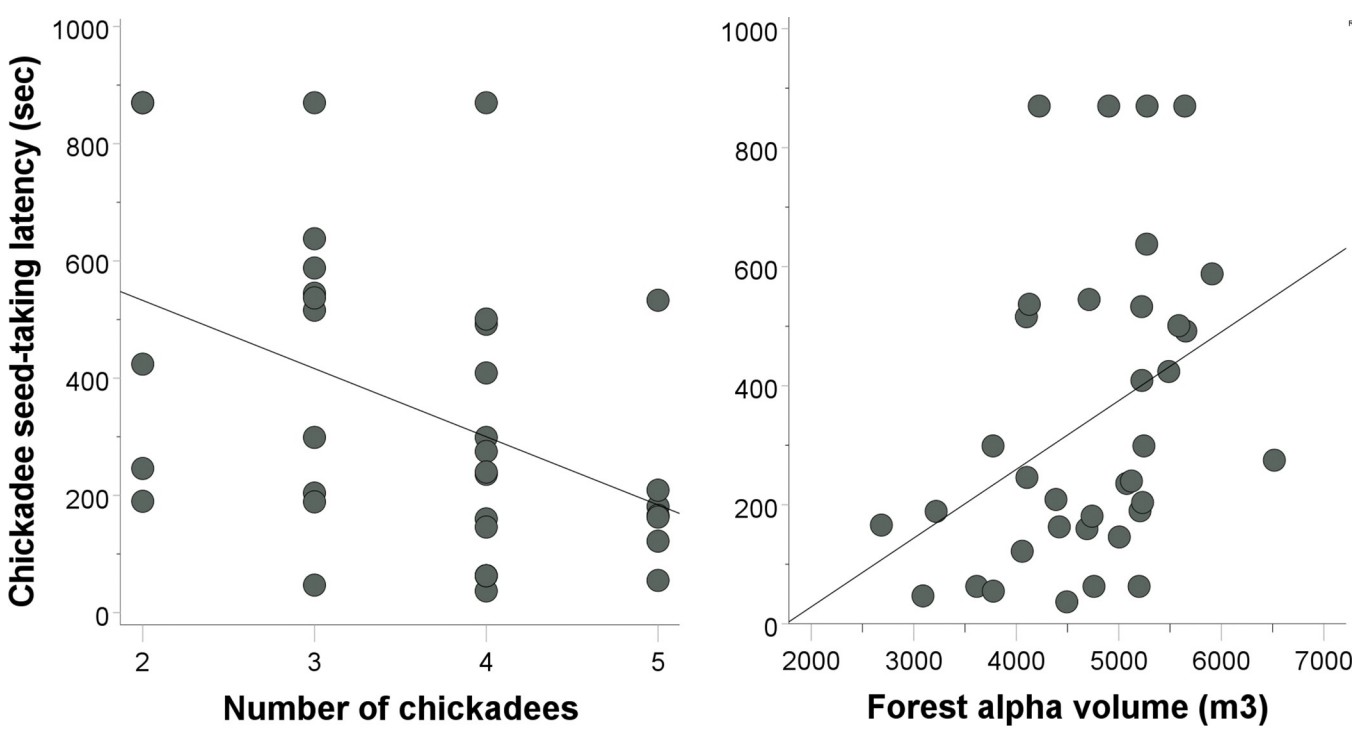

**Fig 1.**  Post-stimulus (alarm playback) chickadee seed-taking latencies in terms of conspecific number (left panel) and vegetation density (forest alpha volume; right panel).

We detected a marginal but not significant effect of conspecific numbers on post-stimulus calling latency for chickadees ($\beta$ = -0.315, t = -1.780, p = 0.086). An individual chickadee tended to call sooner after the playback had ended if there were more chickadees in the flock. None of the other variables predicted chickadee calling latency (flock diversity: $\beta$ = 0.280, t = 1.250, p = 0.221; alpha volume: $\beta$ = 0.146, t = 0.855, p = 0.400; traffic noise: $\beta$ = 0.139, t = 0.801, p = 0.430; and flock size: $\beta$ = -0.174, t = -0.780, p = 0.442).

**Titmouse seed-taking and calling latency.** We detected a significant effect of flock composition metrics on post-stimulus calling latency for titmice (flock diversity: $\beta$ = 0.655, t = 2.576, p = 0.015, $r^2$ = 0.056; flock size: $\beta$ = -0.684, t = -2.079, p = 0.047, $r^2$ = 0.026). An individual titmouse was more likely to call sooner after the playback had ended in flocks with lower mixed-species diversity (Fig 2, left panel) and if there were more total birds in the flock (Fig 2, right panel). We did not detect an effect of alpha volume ($\beta$ = 0.201, t = 1.224, p = 0.231), number of titmice ($\beta$ = 0.235, t = 0.926, P = 0.362), or traffic noise ($\beta$ = 0.116, t = 0.690, p = 0.496) on titmouse calling latency.

For titmice, we detected no effect of forest alpha volume ($\beta$ = 0.262, t = 1.609, p = 0.118), flock size ($\beta$ = -0.486, t = -1.498, p = 0.145), flock diversity ($\beta$ = 0.297, t = 1.193, p = 0.242), traffic noise ($\beta$ = 0.010, t = 0.058, p = 0.954), or conspecific number ($\beta$ = -0.011, t = -0.045, p = 0.965) on seed-taking latency.

**Nuthatch seed-taking and calling latency.** We detected a significant effect of forest alpha volume on post-stimulus seed-taking latency for nuthatches ($\beta$ = 0.568, t = 3.086, p = 0.005, $r^2$ = 0.208). An individual nuthatch was more likely to take a first seed with a longer delay after the playback had ended if there was greater vegetation density around the feeder (Fig 3). We were unable to detect an effect of traffic noise ($\beta$ = 0.211, t = 1.085, p = 0.289), flock diversity ($\beta$ = -0.200, t = -0.993, p = 0.331), conspecific number ($\beta$ = -0.103, t = -0.537, p = 0.596), or flock

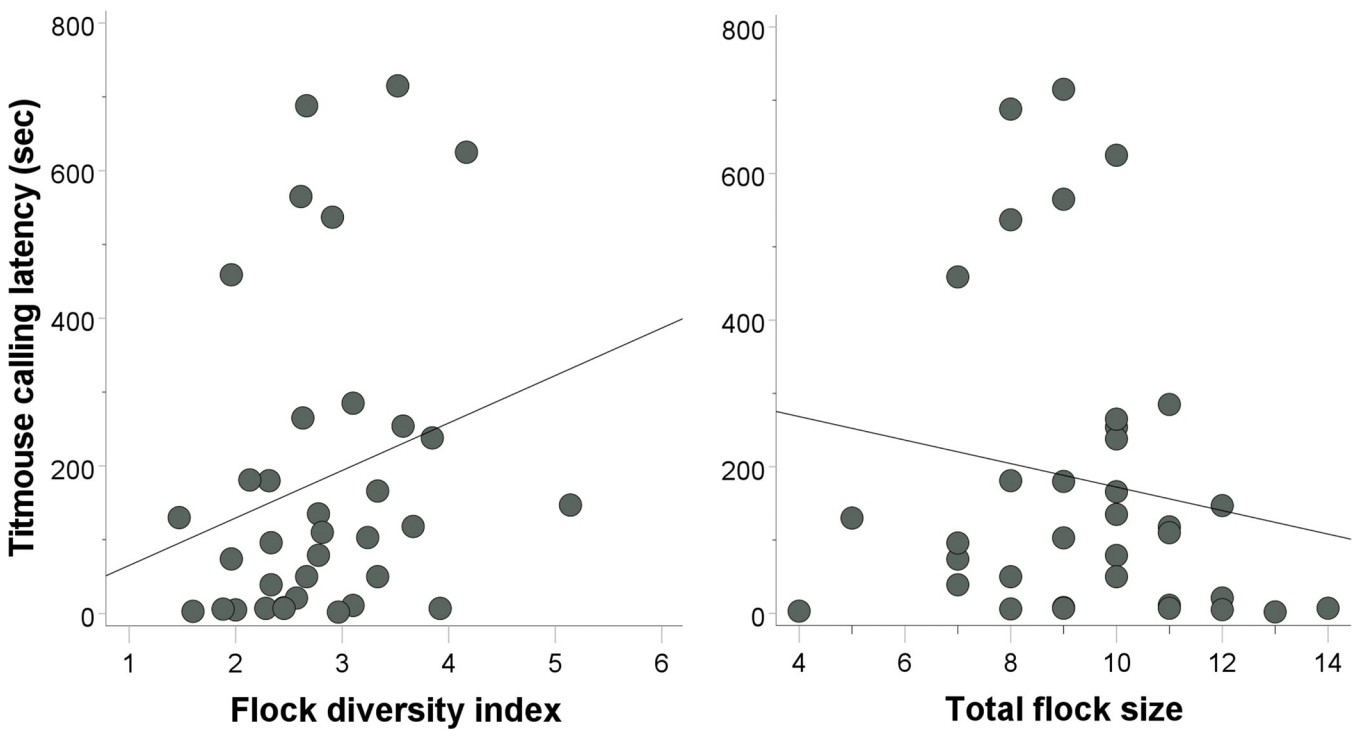

**Fig 2.** Post-stimulus (alarm playback) titmouse calling latencies in terms of mixed-species flock diversity (left panel) and size (right panel).

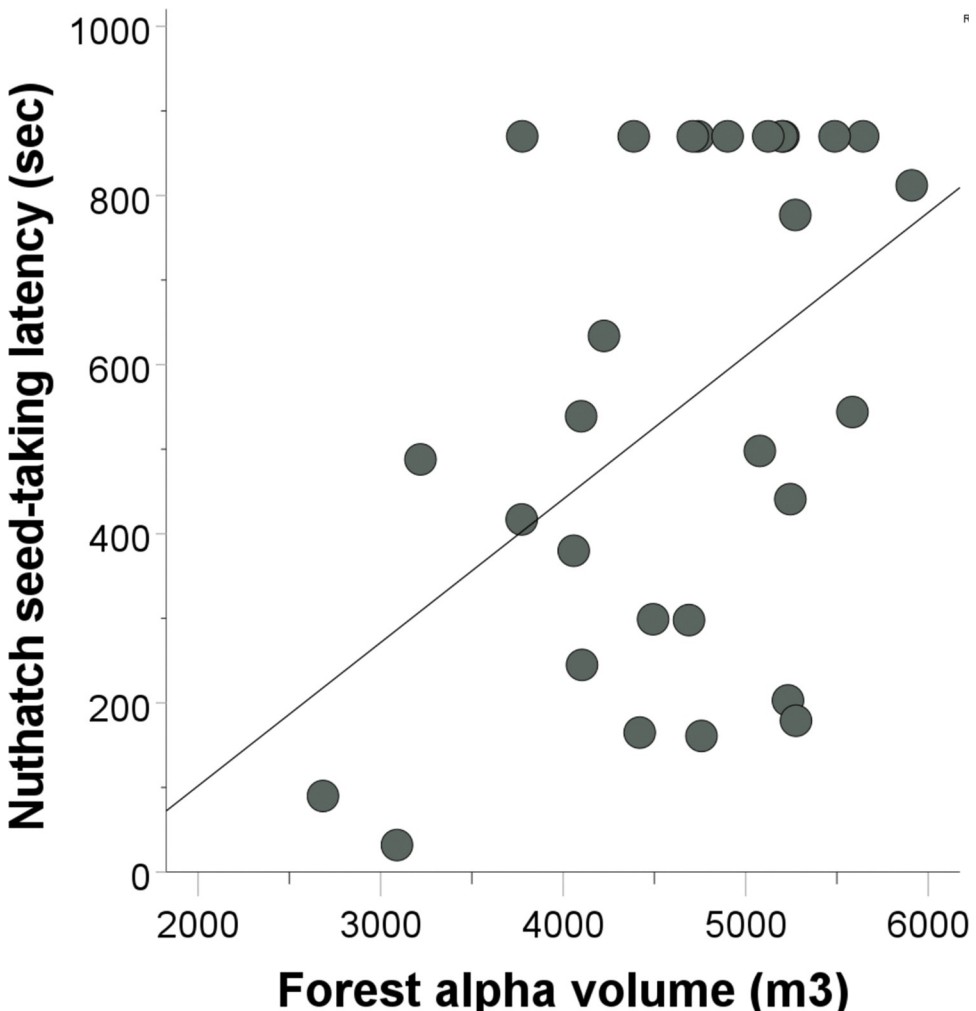

**Fig 3. Post-stimulus (alarm playback) nuthatch seed-taking latencies in terms of vegetation density (forest alpha volume).**

size (β = -0.016, t = -0.084, p = 0.933) on nuthatch seed-taking latency. For nuthatch calling latency, we detected a marginal effect of flock size–nuthatches tended to call earlier the larger their flock was (β = -0.408, t = -2.017, p = 0.056, $r^2$ = 0.090). We detected no effect on calling latency for forest alpha volume (β = 0.271, t = 1.355, p = 0.189), flock diversity (β = 0.262, t = 1.220, p = 0.235), traffic noise (β = 0.235, t = 1.117, p = 0.276), or conspecific number (β = 0.093, t = 0.455, p = 0.653).

**Mobbing duration and mobbing latency.** For the latency to mob following the alarm call playback, we detected a marginal effect of flock size (β = -2.055, t = -2.003, p = 0.056, $r^2$ = 0.100) and of flock diversity (β = 1.081, t = 1.788, p = 0.085)–flock mobbing tended to begin earlier the larger the flock size and the less diverse the flock species composition. We detected no effect of number of chickadees (β = 0.820, t = 1.588, p = 0.124), number of titmice (β = 1.168, t = 1.514, p = 0.142), number of nuthatches (β = 0.422, t = 1.455, p = 0.158), forest alpha volume (β = 0.265, t = 1.542, p = 0.135), or traffic noise (β = 0.052, t = 0.303, p = 0.764) on flock mobbing latency. Flock mobbing duration was also not predicted by any of our social or physical environmental variables (smallest p value for number of nuthatches: β = -0.369, t = -1.129, p = 0.269).

### Predator model presentation study

We observed a total of 138 Carolina chickadees, 109 tufted titmice, and 50 white-breasted nuthatches across the 36 study sites. Carolina chickadees and tufted titmice were present across all 36 sites, whereas white-breasted nuthatches were present for 25 sites. Flocks averaged 3.83 ± 0.24 (mean ± SE) Carolina chickadees, 3.03 ± 0.25 tufted titmice, and 1.39 ± 0.16 white-breasted nuthatches. The other species observed at the feeders included red-bellied (21 individuals across 18 sites) and downy (14 across 14 sites) woodpeckers, Northern cardinals (7 across 4 sites), Carolina wrens (3 across 2 sites), and blue jays (*Cyanocitta cristata*; 5 across 4 sites). The total number of birds observed was 346, the total flock size averaged 9.61 ± 0.46 individuals, and the average flock diversity index was 2.98 ± 0.82.

**Chickadee calling latency.** We detected a significant effect of conspecific number on post-stimulus calling latency for chickadees ($\beta$ = -0.460, t = -2.315, p = 0.028, $r^2$ = 0.156). Chickadees had shorter call latencies as conspecific numbers increased (Fig 4). We detected no effect of forest alpha volume ($\beta$ = -0.261, t = -1.599, p = 0.120), flock diversity ($\beta$ = -0.298, t = -1.308, p = 0.201), traffic noise ($\beta$ = -0.072, t = -0.457, p = 0.651), or flock size ($\beta$ = 0.025, t = 0.095, p = 0.925) on chickadee calling latency.

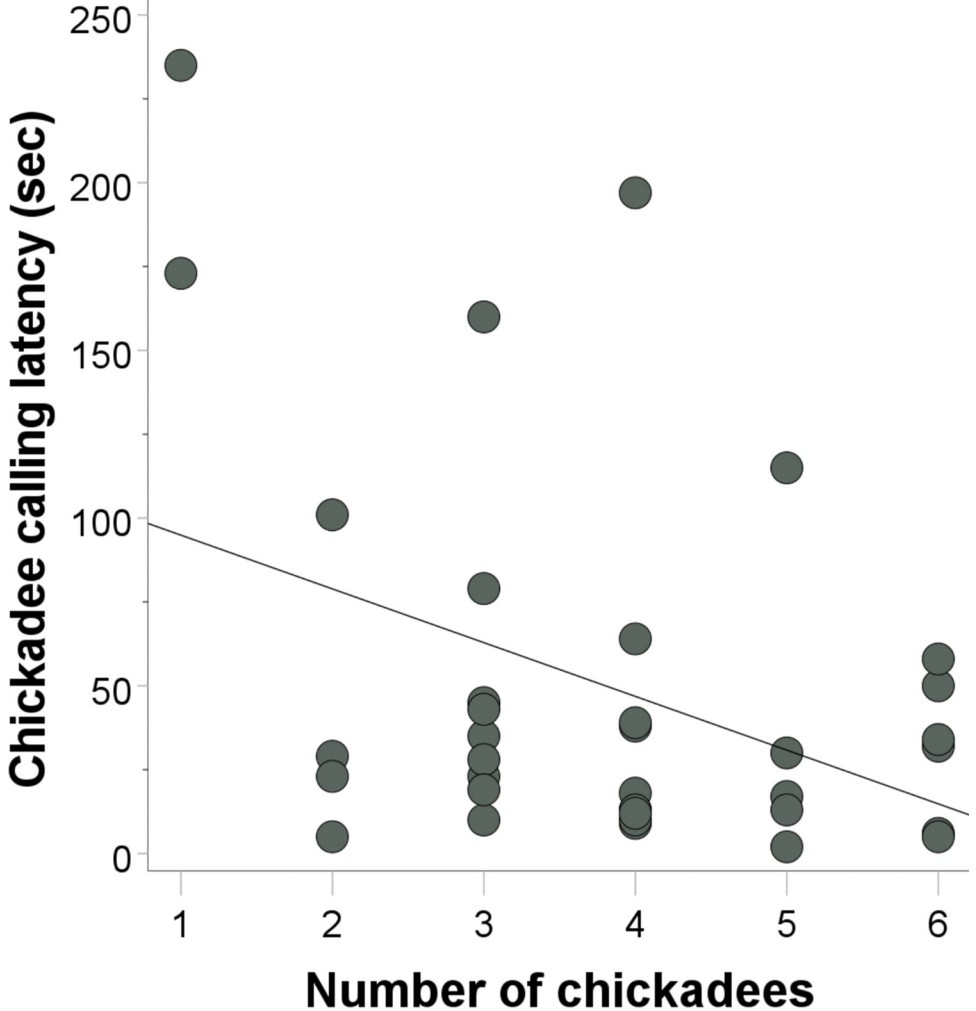

**Fig 4. Post-stimulus (predator model) calling latencies in terms of conspecific number.**

**Titmouse and nuthatch calling latency.** For titmice and for nuthatches, we detected no effect of our five variables on calling latency. The smallest p value for titmice was for flock diversity (β = 0.214, t = 0.848, p = 0.403). The smallest p value for nuthatches was for forest alpha volume (β = 0.188, t = 0.804, p = 0.431).

**Mobbing duration and mobbing latency.** We detected a significant effect of the number of white-breasted nuthatches on post-stimulus flock mobbing duration (β = 0.507, t = 2.442, p = 0.021, $r^2$ = 0.147). Mobbing duration increased as nuthatch numbers increased (Fig 5). Additionally, we detected a tendency for background noise to affect post-stimulus flock mobbing duration (β = -0.306, t = -1.906, p = 0.067). Mobbing duration tended to decrease as background noise intensity increased. We detected no effect on mobbing duration of number of titmice (β = -0.236, t = -0.502, p = 0.620), number of chickadees (β = -0.060, t = -0.128, p = 0.899), flock size (β = -0.161, t = -0.215, p = 0.832), flock diversity (β = 0.064, t = 0.153, p = 0.880), or forest alpha volume (β = -0.025, t = -0.141, p = 0.889). For latency to onset of mobbing, we detected no effect of any of our social or physical environmental variables (smallest p value for forest alpha volume β = -0.295, t = -1.485, p = 0.149).

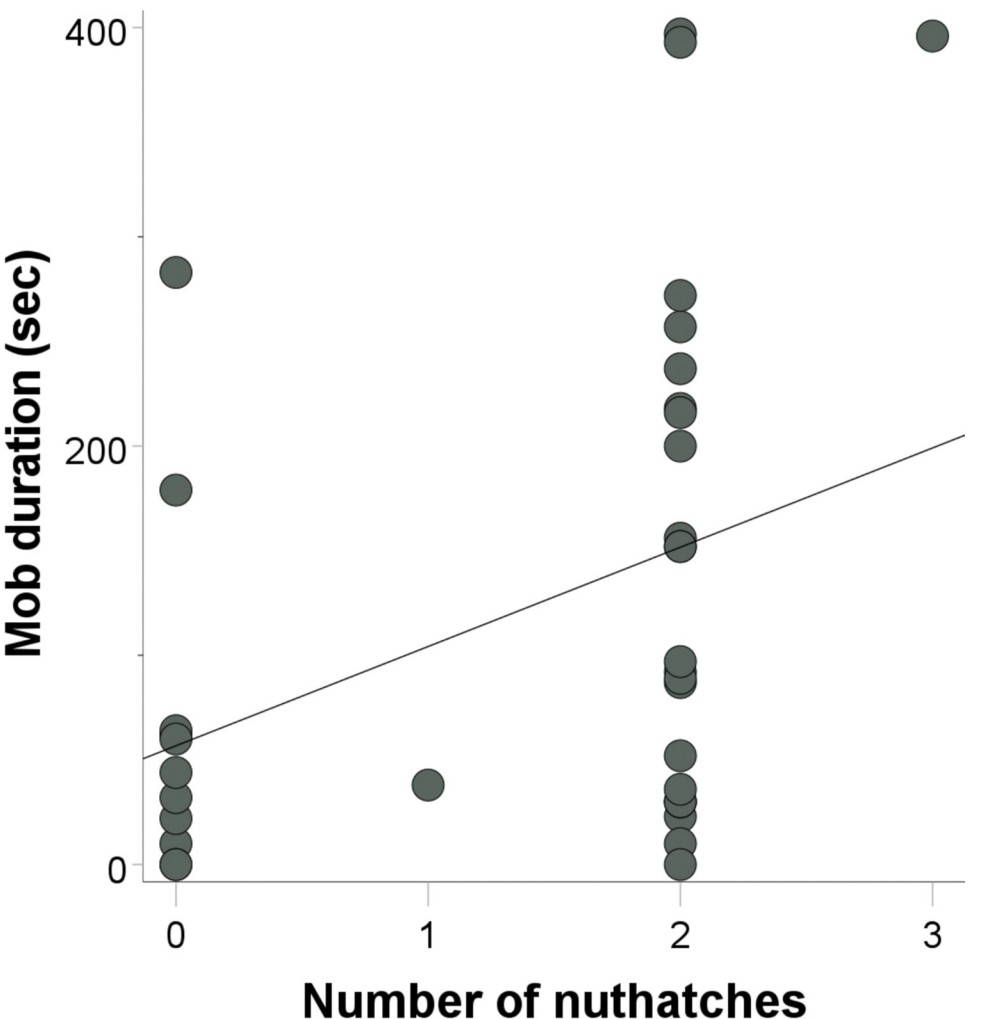

**Fig 5. Post-stimulus (predator model) flock mobbing duration in terms of nuthatch numbers.**

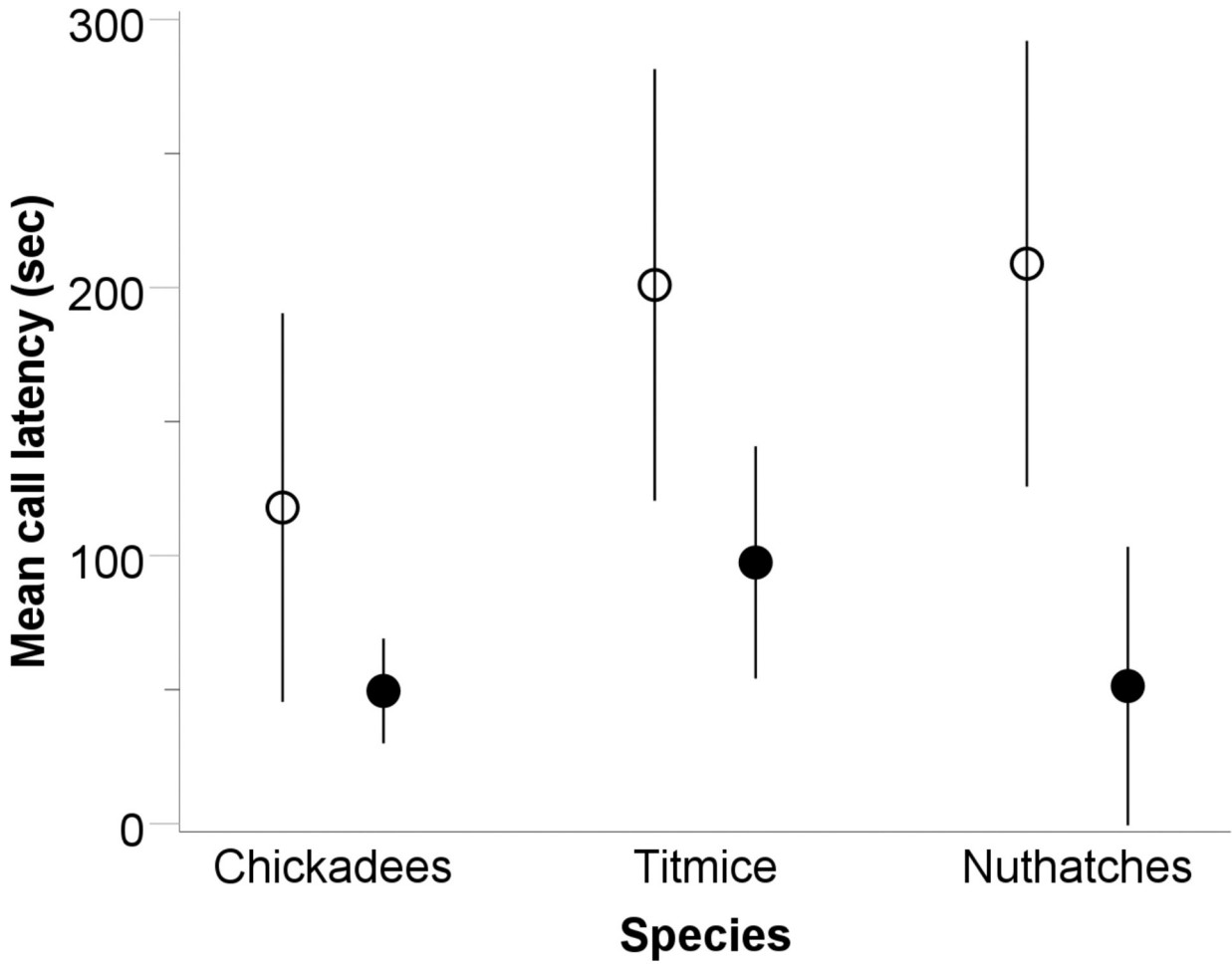

**Fig 6. Post-stimulus mean call latency for species in terms of stimulus modality.** Open circles = alarm call playback study; dark circles = owl model presentation study. Data are plotted as means (circles) and 95% confidence intervals (whiskers). Significant differences were detected for titmice and nuthatches (see text).

### Cross-experiment findings

We detected a significant effect of stimulus modality on post-stimulus calling latencies for titmice (df = 35, t = 2.352, p = 0.024) and nuthatches (df = 19, t = 2.737, p = 0.019). Individual titmice and nuthatches were each more likely to call sooner after the presentation of the owl model than after the alarm call playback (Fig 6). We detected no significant effect of stimulus modality on post-stimulus calling latency for chickadees (df = 35, t = 1.753, p = 0.088), mob latency (df = 33, t = 1.736, p = 0.092), or mob duration (df = 33, t = 0.122, p = 0.903).

### Discussion

In general, our study found that for chickadees, conspecific number was a strong driver of antipredator behavior (H1). In our alarm call study, we found that seed-taking latency decreased if there were more chickadees in a flock, and there was a similar trend for latency to call. In our owl model study, we observed a similar finding for chickadees, where call latency decreased as conspecific numbers increased. Taking these two studies together, chickadees have a tendency to behave more boldly in a riskier context if there are more conspecifics in

their social group. Similar effects of increased conspecific numbers driving stronger anti-predatory behavior have been observed in other species [44, 45]. Familiarity and similarity appear to influence risk assessment and antipredator behaviors in chickadees. Relating to the flock diversity hypothesis (H2), it appeared titmice were influenced more strongly by additional flock composition metrics and not conspecific numbers–in the alarm call study titmice called with a shorter latency in flocks that were large and less diverse in species composition. In the presence of a diverse set of species exhibiting mobbing in their mixed-species flocks, titmice may generally delay or diminish their anti-predatory behavior [21].

Nuthatches in these two studies were almost always found as a pair or were not present in our flocks. As such, conspecific number is likely not a strong predictor of nuthatch behavior. Nuthatch seed-taking behavior in the alarm call study did seem to be influenced by forest alpha volume–nuthatches returned to taking seed more quickly the less dense the nearby vegetation was (the smaller the alpha volume).

The benefits of diversity in mixed-species flocks observed in a novel feeder context [51] may not extend to contexts involving potential predators. The oddity effect suggests that individuals phenotypically different from others in their group may be at greater predation risk [46, 61, 62]. In a more diverse flock, an individual titmouse may stand out more if it were to call earlier. Titmice assuming a passive role in the mixed-species flock system has also been documented in other populations [21, 63], supporting our finding that they are less likely to call sooner after an alarm call presentation. Larger group sizes have been shown to decrease predation risk in mixed-species flocks [64–66]. More individuals within a group make it more difficult to be singled out by a predator [67] and make it easier to spot a predator, as supported by the many-eyes hypothesis [68, 69]. Relating to H1, it appears chickadees are influenced more strongly by conspecific numbers and not additional flock composition metrics. Nuthatches did not seem to attend to flock size or diversity. Though they are common to find within flocks, they are routinely categorized as satellite species within this mixed-species flock system [70]. It would appear that their behavior is driven on an individual level, removed from attention to the diversity within a flock or a flock's size.

In our owl model study, we observed that mob duration increased with nuthatch number in our flocks. Longer mobs are attributed to more aggressive flocks [71]. Smaller body size in mixed-species flocks has been suggested to be associated with greater predation risk [72], so larger-bodied nuthatches may behave bolder if their predation risk is lower in large flocks of smaller species like chickadees. Our results show that the presence of a species with a larger body size and more tendency towards aggression increases overall flock boldness and aggression in a risky situation. Mobbing characteristics were not consistently predicted by flock composition metrics between our studies. The latency and duration of a mob is likely predicted by a variety of confounding factors across both the social and physical environment that these flocks persist in.

As briefly mentioned above, vegetation density also served as a driver of antipredator behavior (H5). In our alarm call study, we found that chickadees and nuthatches were more hesitant to return to the feeder and resume normal seed-taking behavior after the alarm call playback if vegetation near the feeder was denser. The simulated alarm call did not provide individuals in flocks with a visual cue of the potential predator, making it difficult for birds to assess when the period of risk was over. In other words, birds were exhibiting appropriate antipredator behavior under the assumption that a predator was still near. In an environment that has greater vegetation density, furthermore, birds would be more obscured by vegetation and, therefore, less conspicuous to the predator that was assumed to be close. By staying frozen and hidden within the vegetation, birds likely had lower risk of being preyed upon as movement towards the feeder could potentially draw the attention of a predator. However, during the owl

model study, vegetation density did not seem to influence the antipredator behavior of individual birds across our three focal species, nor did vegetation density predict flock mobbing characteristics. When a threat is clearly visible to individuals within a group, vegetation may not impact antipredator behavior because birds are not as reliant on it for cover, as appears the case when the predation cue is acoustic.

Background noise intensity surprisingly played little role in antipredator behavior in our two studies (H4). In our owl model study, we observed that mob duration tended to decrease with increasing levels of background noise, though the result was not statistically significant. Mobbing is energetically expensive and puts individuals at risk of direct predation [73, 74]. In the case of our owl model study, birds could directly see the predator model on the feeder. Given that the function of mobbing is to intimidate and potentially drive away a predator, a mob would be most effective in an environment where individuals would be able to clearly detect these calls from other mobbers, signaling them to join and contribute. In a noisier environment, potential mobbers would be less likely to detect and clearly discriminate between mob calls and background noise, so mobbing for a longer period of time could be an ineffective antipredator strategy. This supports the masking hypothesis, whereby call detection is decreased as a function of increasing background noise intensity [26, 27, 75]. Our results suggest that individuals within social groups are perceptive of their environment's soundscape and of a potential predator's actions, and they alter their antipredator responses accordingly, so as to not waste energy and be positioned in potentially risky contexts for longer than necessary. During the alarm playback study, though, background noise did not predict antipredator behavior across any of the three focal species, nor did it drive the mobbing characteristics of flocks. Given the high frequency of the notes in alarm calls of titmice, including our playback stimuli, it seems likely that these calls would have been perceived by the birds even at the highest traffic noise sites, as traffic noise typically exists at a low range of frequencies [76]

Finally, simulated predation risk modality influenced antipredator behavior differences. When comparing the averages of our measured antipredator behaviors between the alarm and owl studies, we found that calling latency was shorter for titmice and nuthatches if the stimulus presented was visual. The visual cue of the owl on the feeder was assessed as an immediate predation risk by individuals, resulting in quicker call elicitation following this stimulus presentation than the acoustic playback we used to simulate predation risk. If individual birds in flocks can visually locate a predator, there is no uncertainty about the nature of the risk. In the alarm call experiment, however, birds had no visual cue of where a predator might be and so uncertainty remained. Calling in this latter case would therefore likely be riskier, resulting in an increased time of silence following the acoustic predation cue. These results support previous studies [77–79] that found if a predator stimulus is visual as opposed to acoustic, then foraging rates in those prey species are reduced and anti-predatory behavior (vigilance, calling) is increased. Interestingly, with the exception of chickadee calling latency being predicted by conspecific number, our results are not mirrored between the studies. For example, titmice appeared to attend to flock size and flock diversity for calling latency in our alarm study, but not in our owl study. Similarly, mobbing duration was predicted by two factors in our owl model study, but nothing predicted it in our alarm call study. This discrepancy in and of itself demonstrates how modality of predation risk assessment can drive differences in antipredator behaviors. More work is needed in these mixed-species flocks to clarify whether the different effects of modality that we found here are due to the alarm study representing a moving predator context [52] or a context of heightened uncertainty, compared to the perched visual stimulus of the owl model.

## Conclusions

Overall, our results support the idea that individuals pay attention to conspecific and hetero-specific group composition. Chickadees and titmice tend to behave more boldly in a riskier context if there are more conspecifics in a group. Titmice furthermore behave bolder in a riskier context if there are more total individuals in their mixed-species flock. Flocks have a tendency to behave more boldly in a riskier context if larger, more aggressive nuthatches are present in a group. These findings point to how sensitive individuals are to variation in their current environmental context. The sensitivities of individuals to their social environment in the context of foraging and calling behavior under risk is highlighted by our results, especially in regard to nuclear and satellite species attending to the relative proportions of one another within mixed-species groups.

Moreover, one of the two variables we assessed relating to environmental noise (vegetation density) influenced the antipredator behaviors of individuals. Our study is among the first to use terrestrial lidar data to quantify vegetation density as a potential predictor variable in studying the behavior of animals. We found that individual foraging behavior is affected by density of vegetation when in a risky context of alarm calling–birds took longer to resume foraging in denser habitat [see also 80, 81]. Future studies could seek to implement data extracted from terrestrial lidar scans as a predictor for the behavioral responses of visually and acoustically reliant animals under a variety of social contexts, including risk assessment, foraging, communication, and reproduction.

## Supporting information

**S1 Fig. A 10 sec clip of the spectrogram output for the 60 sec alarm call playback we used.** (TIF)

**S2 Fig. Output for average background noise level across our sites.** (TIF)

**S1 File. Our owl model on a feeder stocked with seed.** (TIF)

## Acknowledgments

The authors thank the staff at UTFRREC for allowing us to carry out our study on their grounds. We thank the National Institute for Mathematical and Biological Synthesis and the Spatial Analysis Lab at the University of Tennessee, Knoxville (UTK), as well as Dr. Árpád S. Nyári, for lending us their resources and the lidar scanner. We thank Alexander Miele and Mark Young for help with the construction of the site locations map. We thank Scott Benson, Heather Brooks, Eric Frazier, Harry Pepper, and Ryan Risner for helpful critiques of earlier drafts of this manuscript. We thank the Center for the Dynamics of Social Complexity, the Department of Psychology, and the College of Arts and Science at UTK for funding that helped make the study possible.

## Author Contributions

**Conceptualization:** Colton B. Adams, Monica Papeş, Todd M. Freeberg.

**Data curation:** Colton B. Adams, Monica Papeş, Charles A. Price, Todd M. Freeberg.

**Formal analysis:** Monica Papeş, Charles A. Price, Todd M. Freeberg.

**Funding acquisition:** Monica Papeş, Todd M. Freeberg.

**Investigation:** Colton B. Adams, Monica Papeş, Charles A. Price.

**Methodology:** Colton B. Adams, Monica Papeş, Charles A. Price.

**Project administration:** Todd M. Freeberg.

**Resources:** Monica Papeş, Todd M. Freeberg.

**Software:** Monica Papeş, Charles A. Price, Todd M. Freeberg.

**Supervision:** Monica Papeş, Charles A. Price, Todd M. Freeberg.

**Validation:** Colton B. Adams, Monica Papeş, Charles A. Price, Todd M. Freeberg.

**Visualization:** Colton B. Adams, Todd M. Freeberg.

**Writing – original draft:** Colton B. Adams.

**Writing – review & editing:** Colton B. Adams, Monica Papeş, Charles A. Price, Todd M. Freeberg.

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
