## [Decision Letter · Decision Letter 0]

16 Oct 2023

PONE-D-23-25551Influence of social and physical environmental variation on antipredator behavior in mixed-species flocksPLOS ONE

Dear Dr. Freeberg,

Thank you for submitting your manuscript to PLOS ONE. After careful consideration, we feel that it has merit but does not fully meet PLOS ONE’s publication criteria as it currently stands. Therefore, we invite you to submit a revised version of the manuscript that addresses the points raised during the review process.

Both reviewers clearly agree this is a well done, important piece of research. The minor revisions largely need to focus on greater clarity on some points as they point out.

We look forward to receiving your revised manuscript.

Kind regards,

Vivek Nityananda

Academic Editor

PLOS ONE

Journal Requirements:

4. We note that [S1 File: Aerial image of our field site showing feeder distribution.] in your submission contain [map/satellite] images which may be copyrighted. All PLOS content is published under the Creative Commons Attribution License (CC BY 4.0), which means that the manuscript, images, and Supporting Information files will be freely available online, and any third party is permitted to access, download, copy, distribute, and use these materials in any way, even commercially, with proper attribution. For these reasons, we cannot publish previously copyrighted maps or satellite images created using proprietary data, such as Google software (Google Maps, Street View, and Earth). For more information, see our copyright guidelines: http://journals.plos.org/plosone/s/licenses-and-copyright.

a. You may seek permission from the original copyright holder of S1 File: Aerial image of our field site showing feeder distribution. to publish the content specifically under the CC BY 4.0 license.  

Reviewers' comments:

Reviewer's Responses to Questions

**Comments to the Author**

1. Is the manuscript technically sound, and do the data support the conclusions?

Reviewer #1: Yes

Reviewer #2: Yes

2. Has the statistical analysis been performed appropriately and rigorously? 

Reviewer #1: Yes

Reviewer #2: Yes

3. Have the authors made all data underlying the findings in their manuscript fully available?

Reviewer #1: Yes

Reviewer #2: Yes

4. Is the manuscript presented in an intelligible fashion and written in standard English?

Reviewer #1: Yes

Reviewer #2: Yes

5. Review Comments to the Author

Reviewer #1: This is an interesting and well replicated set of experiments that further demonstrate the extreme sensitivity of birds to their social context, both inter and intraspecific. The writing is also clear and easy to follow with the exception of a few minor issues described below.

I was mildly surprised to see that the effects of noise were so small, especially given the wide range of background levels available. I did not understand the reasoning in lines 495-497 to explain these generally milder responses than expected. This sentence needs rewording.

I also wonder if calling the screech-owl a predator of adult flockers is legitimate. The reference (39) was garbled, so I could not check it out. This species spends the daylight hours in cavities where it would presumably be of little danger. Rather, it seems likely that it would be a threat to birds in their nocturnal roosts and therefore worth chasing away. This might require some slight rewording as the threat was perhaps not immediate as it would have been with a model Accipiter. The milder response to this species may reflect that birds do not regard this as a serious immediate threat.

36 feeding stations is an impressive number for this sort of experiment; do you have any evidence that the same individuals were using more than one feeding station? More information on the spacing would help, although the supplemental figure shows the distribution well, it does not provide distance estimated.

Reviewer #2: I would like to congratulate the authors on this manuscript, it is well conceptualised with very clear statement and testing of hypotheses. The field experiments conducted are also well thought off and the authors have attempted to be as thorough with the design. The manuscript is well written, and the language is clear and easy to follow. I feel the authors may need to think a little bit more about the result and work on the discussion before the manuscript can be published. Below are a few suggestions and clarifications.

Line 135: Regarding hypothesis 4, I was wondering if increased vegetation reduces propagation of sound? Which would reduce the effectiveness of mobbing. Also increased vegetation would make the predator also be less conspicuous right which should then make birds more cautious thus increasing the latencies of behaviour?

Line 171: Typo, should be “playback”

Line 209: I don’t know if doing an additional experiment is possible, but I was wondering what would happen if you placed a large non predatory bird like a duck/goose on the platform. Through the current experimental set-up it’s a bit difficult to claim that the tits and nuthatches are actually recognising the owl or is it just an unfamiliar object so close to its feeding station.

Line 240-241: Useful to mention how "density may influence anti-predatory behaviors in our mixed-species flock system"

Line 263: Why remove variables I feel it would be better to just interpret the global model

Line 370-371: If the owl is still visible why would the chikadee resume calling?

Line 425-426: By other species you mean from other studies?

Line 427-428: Suggest rephrasing because you speak about the H2 result in more detail in the next paragraph

Line 429: Replace phrases “not at all” and “solid” to make sentence sound more formal

Line 436-468: Sentence is a little confusing, if the authors are referring to another study, make it more explicit

Line 452-453: Sentence needs reference

Line 495-497: The authors may need to explain this sentence a little more because earlier you find that due to background noise, birds are possibly unable to discriminate mob calls but that doesn't seem to be the case with alarm calls

Line 503-505: Ambush yes, but they are still at risk as calling makes them more visible to the predator. I don't think the author's have sufficiently explained why latency is longer when the cue is aural. I would assume call latency in the case of the visual cue would continue until the visual cue is no longer present.

Line 508-509: Not clear what “foraging will be less likely” and “antipredator calling more likely” means

Line 514-515: This is ok, but I feel the authors need to reflect a little more about these discrepancies

Line 535-536: Consider rephrasing this sentence as it is a little unclear “We found that individuals likely attend to vegetation cover when under a risky context”

Line 537: Slightly unclear what the authors mean by “decreased foraging behaviour”

6. PLOS authors have the option to publish the peer review history of their article (what does this mean?). If published, this will include your full peer review and any attached files.

Reviewer #1: No

Reviewer #2: No

---

## [Author Response · Author response to Decision Letter 0]

27 Nov 2023

Note that the red highlighted text providing our responses to reviewers' comments is lost here - please see the Response to Reviewers file we submitted as the cover letter!

Response to reviewers – PLoSONE ms PONE-D-23-25551

Influence of social and physical environmental variation on antipredator behavior in mixed-species flocks

Dear Dr. Nityananda:

Thank you for giving us the opportunity to revise and resubmit our manuscript to address the helpful criticisms and questions of the reviewers. We have copied and pasted the entire set of comments of the reviewers, which we include below. For each comment, we briefly describe in red text how it was addressed and, if relevant, where the changes can be found (by line number) in the clean, non-track-changes manuscript.

+++++++++++++++++++++++++

Thank you for submitting your manuscript to PLOS ONE. After careful consideration, we feel that it has merit but does not fully meet PLOS ONE’s publication criteria as it currently stands. Therefore, we invite you to submit a revised version of the manuscript that addresses the points raised during the review process.

Both reviewers clearly agree this is a well done, important piece of research. The minor revisions largely need to focus on greater clarity on some points as they point out.

We thank you and the reviewers for the generally positive view of our manuscript!

Reviewers' comments:

5. Review Comments to the Author

Reviewer #1: This is an interesting and well replicated set of experiments that further demonstrate the extreme sensitivity of birds to their social context, both inter and intraspecific. The writing is also clear and easy to follow with the exception of a few minor issues described below.

I was mildly surprised to see that the effects of noise were so small, especially given the wide range of background levels available. I did not understand the reasoning in lines 495-497 to explain these generally milder responses than expected. This sentence needs rewording.

Good point – we have deleted the sentence and added text in lines 521-524 to address the key idea here.

I also wonder if calling the screech-owl a predator of adult flockers is legitimate. The reference (39) was garbled, so I could not check it out. This species spends the daylight hours in cavities where it would presumably be of little danger. Rather, it seems likely that it would be a threat to birds in their nocturnal roosts and therefore worth chasing away. This might require some slight rewording as the threat was perhaps not immediate as it would have been with a model Accipiter. The milder response to this species may reflect that birds do not regard this as a serious immediate threat.

We added some text to address this in lines 97-99. Screech owls are indeed not particularly active in their hunting during the day, chickadees, titmice, and nuthatches respond strongly to screech owl stimuli – both visual and acoustic – during the daylight hours.

36 feeding stations is an impressive number for this sort of experiment; do you have any evidence that the same individuals were using more than one feeding station? More information on the spacing would help, although the supplemental figure shows the distribution well, it does not provide distance estimated.

We added text in lines 152-154 to give better support to the argument that our feeder separation distances represent the sampling of different flocks – this information came from the cited study by Bartmess-LeVasseur et al. (2010).

Reviewer #2: I would like to congratulate the authors on this manuscript, it is well conceptualised with very clear statement and testing of hypotheses. The field experiments conducted are also well thought off and the authors have attempted to be as thorough with the design. The manuscript is well written, and the language is clear and easy to follow. I feel the authors may need to think a little bit more about the result and work on the discussion before the manuscript can be published. Below are a few suggestions and clarifications.

We thank the reviewer for the supportive comments on our study and manuscript!

Line 135: Regarding hypothesis 4, I was wondering if increased vegetation reduces propagation of sound? Which would reduce the effectiveness of mobbing. Also increased vegetation would make the predator also be less conspicuous right which should then make birds more cautious thus increasing the latencies of behaviour?

The densities of the vegetation at our feeding stations sites are not such that they would impact sound transmission at such short distances. But, for sure, the last question raised by the reviewer is spot on, and we realize we incorrectly laid out the predictions of this hypothesis on these lines – very grateful to the reviewer for catching this! Revised text in lines 136-141.

Line 171: Typo, should be “playback”

Although at least one of the authors is a James Brown fan, the reviewer is correct and it should have been ‘playback’ – corrected in line 174!

Line 209: I don’t know if doing an additional experiment is possible, but I was wondering what would happen if you placed a large non predatory bird like a duck/goose on the platform. Through the current experimental set-up it’s a bit difficult to claim that the tits and nuthatches are actually recognising the owl or is it just an unfamiliar object so close to its feeding station.

The reviewer raises an interesting point, and one we cannot address in the present studies. In the 2010 study mentioned above (Bartmess-LeVasseur et al.) we used plastic hawk and owl models set at different distances from the feeding stations. The birds reacted much more strongly to the hawk models (decreased seed-taking, increased calling) than to the dove models.

Line 240-241: Useful to mention how "density may influence anti-predatory behaviors in our mixed-species flock system"

We added text to clarify this in lines 245-248.

Line 263: Why remove variables I feel it would be better to just interpret the global model

We followed the reviewer’s suggestion (and the advice from a textbook on these statistical models) and used just the full regression models with no iteration.

Line 370-371: If the owl is still visible why would the chikadee resume calling?

As we raise in the discussion, we think this is evidence of chickadee mobbing behavior beginning more quickly if there are more chickadees in the flocks.

Line 425-426: By other species you mean from other studies?

We added some text to this sentence to try to make it clearer that this sentence refers to published studies with other species – lines 451-452.

Line 427-428: Suggest rephrasing because you speak about the H2 result in more detail in the next paragraph

Clarified in line 453-458.

Line 429: Replace phrases “not at all” and “solid” to make sentence sound more formal

Done in lines 459-460.

Line 436-468: Sentence is a little confusing, if the authors are referring to another study, make it more explicit

Text modified in this sentence to try to make the point clearer – lines 456-458.

Line 452-453: Sentence needs reference

We thank the reviewer for catching this – we added the classic Morse 1970 reference on mixed-species flocks in line 476.

Line 495-497: The authors may need to explain this sentence a little more because earlier you find that due to background noise, birds are possibly unable to discriminate mob calls but that doesn't seem to be the case with alarm calls

As commented on above to reviewer 1, we revised the text on this point in lines 521-524.

Line 503-505: Ambush yes, but they are still at risk as calling makes them more visible to the predator. I don't think the author's have sufficiently explained why latency is longer when the cue is aural. I would assume call latency in the case of the visual cue would continue until the visual cue is no longer present.

We revised the text in lines xxx-xxx to try to make our argument here clearer.

Line 508-509: Not clear what “foraging will be less likely” and “antipredator calling more likely” means

We revised the text here to try to make our arguments clearer – lines 530-534.

Line 514-515: This is ok, but I feel the authors need to reflect a little more about these discrepancies

We added a concluding sentence to try to beef up this point a bit more – lines 541-546.

Line 535-536: Consider rephrasing this sentence as it is a little unclear “We found that individuals likely attend to vegetation cover when under a risky context”

We modified the text here to try to make the point clearer – lines 562-564.

Line 537: Slightly unclear what the authors mean by “decreased foraging behaviour”

This sentence was deleted given the changes made in light of the previous comment.

---

## [Editor Report · Decision Letter 1]

4 Dec 2023

Influence of social and physical environmental variation on antipredator behavior in mixed-species parid flocks

PONE-D-23-25551R1

Dear Dr. Freeberg,

We’re pleased to inform you that your manuscript has been judged scientifically suitable for publication and will be formally accepted for publication once it meets all outstanding technical requirements.

Kind regards,

Vivek Nityananda

Academic Editor

PLOS ONE
---

## [Editor Report · Acceptance letter]

12 Dec 2023

PONE-D-23-25551R1 

Influence of social and physical environmental variation on antipredator behavior in mixed-species parid flocks 

Dear Dr. Freeberg:

I'm pleased to inform you that your manuscript has been deemed suitable for publication in PLOS ONE. Congratulations! Your manuscript is now with our production department. 

Kind regards, 

on behalf of

Dr. Vivek Nityananda 

Academic Editor

PLOS ONE